# Electrochemical Signal Substance for Multiplexed Immunosensing Interface Construction: A Mini Review

**DOI:** 10.3390/molecules27010267

**Published:** 2022-01-02

**Authors:** Jiejie Feng, Changshun Chu, Zhanfang Ma

**Affiliations:** Department of Chemistry, Capital Normal University, Beijing 100048, China; fengjiejiecnu@126.com (J.F.); ccs_ic@163.com (C.C.)

**Keywords:** electrochemical immunosensing interface, simultaneous analysis for multiple tumor markers, electrochemical signal substance, accurate diagnosis of cancer

## Abstract

Appropriate labeling method of signal substance is necessary for the construction of multiplexed electrochemical immunosensing interface to enhance the specificity for the diagnosis of cancer. So far, various electrochemical substances, including organic molecules, metal ions, metal nanoparticles, Prussian blue, and other methods for an electrochemical signal generation have been successfully applied in multiplexed biosensor designing. However, few works have been reported on the summary of electrochemical signal substance applied in constructing multiplexed immunosensing interface. Herein, according to the classification of labeled electrochemical signal substance, this review has summarized the recent state-of-art development for the designing of electrochemical immunosensing interface for simultaneous detection of multiple tumor markers. After that, the conclusion and prospects for future applications of electrochemical signal substances in multiplexed immunosensors are also discussed. The current review can provide a comprehensive summary of signal substance selection for workers researched in electrochemical sensors, and further, make contributions for the designing of multiplexed electrochemical immunosensing interface with well signal.

## 1. Introduction

Cancer is one of the main causes of human death, and its accurate diagnosis always requires a comprehensive monitor for correlative biomarkers [1,2,3]. However, in the present detection of cancer biomarkers, most immunoassays have focused on the improvement of analysis performance, including the limit of detection, detection range, sensitivity or other parameters, and exploitation of novel immunoassay methodologies to contrapose the analysis of single tumor marker [4,5,6,7]. In fact, most cancers have more than one tumor marker associated with their occurrence, and many markers are correlated to more than one kind of cancer or disease condition [8,9,10]. No single tumor marker can specifically meet the exact diagnosis standard, so that many false negative or positive perceptions have been inevitably caused in the traditional single-target immunoassay [11,12,13]. Hence, for the sake of the accurate diagnosis of cancer, an increasing number of analytical techniques like fluorescence [14,15,16], electrochemiluminescence [17,18,19], enzyme-linked immune-sorbent [20,21,22], electrochemistry [23,24,25], and mass spectrometry [26,27] and some others have been designed to satisfy the need for the simultaneous detection of multifarious tumor markers, which are interrelated to provide more numerical references for one cancer. Among the subsistent detection method for cancer biomarkers detection, the electrochemical method has been successfully developed as a sophisticated tool on account of its highly sensitive, superior selective, fast response, and simple operation, efficaciously achieving the accurate diagnosis of cancer [28,29,30,31].

Sensitivity, as the crucial parameter for the evaluation of an electrochemical immunosnesor, is always defined as the signal variation caused by the incubation of antigen per unit concentration [32,33,34,35]. High sensitivity is beneficial for the accurate detection of multiple tumor markers [36,37,38]. Labeling of electrochemical signal substance, a necessary step in the construction of electrochemical immunosening interface [39,40], is closely associated with the final readout of electrochemical signal to influence the sensitivity of immunosensor [41,42]. However, to simultaneously determine the content of multiple tumor markers with high sensitivity, it may demand the labeling of multiple sorts of signal substance or construct one signal-labeled pervasive immunoassay array so that the requirements for the electrochemical signal substance always need convenient loading, easy availability, and excellent electro-redox characterizations [43,44]. Hence, the selection of signal substance is vital to construct an electrochemical immunosensing interface for the analysis of multiple tumor markers.

In this review, we have condensed on the recent advances for the application of an electrochemical signal substance to construct multiplexed electrochemical immunosensing interface, including the usage of dyestuff molecules (methylene blue (MB), thionine (THI)), metal ions (copper(II) ions, lead(II) ions, titanium(IV) ions, cadmium(II) ions, and zinc(II) ions), metal nanoparticles (silver nanoparticle and copper nanoparticle), and other synthesized electroactive materials like Prussian blue. Besides, the current challenges and future prospects are also summarized and discussed. This timely review is aimed to provide a table of reported signal substances used in multiplexed electrochemical immunosensor and help researchers design excellent immunosensing interfaces with good signaling in the future.

## 2. Electrochemical Signal Substances for Multiplexed Immunoassay

### 2.1. Organic Molecules

Organic molecules, as the most common redox species, often serve as the electrochemical signal substance to construct the immunosensing interface [45,46]. Among the organic molecules, dyestuff like MB [47,48] and THI [49,50], and other organic redox compounds like ferrocene, are the most representative on account of their easy accessibility, simple loading method, and excellent redox spikes [51,52]. Kong et al. once used MB and THI as the electroactive labels to construct a multiplexed electrochemical immunosensing interface for the simultaneous detection of carcinoembryonic antigen (CEA) and α-fetoprotein (AFP) [53]. They have employed the carboxyl graphene nanosheets (GS) as the substrate materials to load MB and THI molecules, respectively, finally preparing the carboxyl graphene nanosheets-methylene blue (CGS-MB) and carboxyl graphene nanosheets-Prussian Blue (CGS-PB) nanocomposites. For the subsequent process of immune incubation, the carboxyl groups on the GS surface were used as the anchors to fix the coating antibody (Ab_1_), finally constructing such two sorts of immunosensing interfaces. Meanwhile, these two peaks (related to the reduction of PB and MB) were posited at +0.24 V and −0.33 V after a Differential Pulse Voltammetry (DPV) test, indicating the content of AFP and CEA, respectively. With this method, this immunosensor exhibited an excellent detection performance (a wide detection range of 0.8–80 ng mL^−1^ for CEA and 0.5–50 ng mL^−1^ for AFP). 

Besides the labeling of methylene blue molecules, Alizadeh et al. reported a sandwich-type electrochemical immunoassay based on two categories of multi-functionalized gold nanoparticles (Au NPs) for simultaneous determination of CEA and AFP (Figure 1(1)) [54]. They fabricated this sensor in virtue of co-immobilization of coating anti-CEA and labeling anti-AFP antibodies on the surface of Fe_3_O_4_ NPs and prepared two tags, including thionine-Au NPs and ferrocene-Au NPs employed as the distinguishable labels to indicate the content of CEA and AFP, respectively (Figure 1(3)). Then, in order to testify the feasibility of this experimental mechanism, a series of different concentrations for biomarkers were incubated, and hence corresponding current responses were gradually increased, as shown in Figure 1(2). 

Another interesting example reported the four kinds of redox bioprobes as the tool for simultaneously detecting CEA, carbohydrate antigen (CA) 19-9, 12-5, and 24-2 [55]. With the marking of four electroactive species including anthraquinone 2-carboxylic acid, THI, tris(2,2(-bipyridine-4,4(-dicarboxylic acid) cobalt(III) and ferrocenecarboxylic acid, labeling antibodies (Ab_2_) for CEA, CA 19-9, CA 12-5, and CA 24-2 can be efficaciously labeled to prepare four redox substance@Ab_2_ bioconjugates, and subsequently these bioconjugates were composited with carbon nanotubes (CNTs), which were linked with poly(diallyldimethylammonium chloride) (PDDA), 3,4,9,10-perylenetetracarboxylic acid (PTCA) and AuNPs, to form the final functional immunoprobes. Besides the above design, the glass carbon electrode (GCE) modified by Au NPs was adopted as an ordinary substrate to link with the Ab_1_. Once the sandwich bioconjugates were formed and readout in an electrolyte solution, DPV scan exhibited four well-resolved peaks to indicate the content of responsive antigen (Peaks posited at −0.52V, −0.21V, 0.0V, and 0.26V were corresponding to the CEA, CA 19-9, CA 12-5, and CA 24-2). Importantly, various ingredients in bioprobes can greatly accelerate the electron transfer, and the effect on DPV responses was verified, revealing that the current response was gradually increased as the assembly of Au NPs, PDDA/CNTs, and PTCA in turn. In addition, the cross-reactivity for the immunoassay was also verified to be well by testing the effects of coexistent analytes on the signal for each immunoreaction, the changes of the current response for CEA, CA 19-9, CA 12-5, and CA 72-4 were less than 1.3%, 2.1%, 2.6%, and 1.4%, respectively. Non-specific interference (Bovine Serum Albumin (BSA), AFP, and prostate-specific antigen (PSA)) test was also finally tested to be excellent by adding various noncognate proteins to mixed target analytes. With the assistance of this designing strategy, the proposed multiple immunosensor displayed a series of wide detection ranges: 0.016 to 15 ng mL-1 for CEA, 0.008 to 10 ng mL^−1^ for CA 19-9, 0.012 to 12 ng mL^−1^ for CA 12-5, and 0.01 to 10 ng mL^−1^ for CA 24-2.

Based on the labeling of organic molecules, the multiplexed immunosensor all exhibited considerable electrochemical signals, which can be independently present at the corresponding position. Especially for the organic dyestuff, they can generate a high current value even if it was the trace.

### 2.2. Metal Ions

Metal ions possess good electrochemical redox activity, which makes them ideal materials as electrochemical signal substances [56,57]. With the advantages of easy accessibility and generated sharp electrochemical signal peaks, metal ions like silver ions, copper(II) ions, lead(II) ions usually serve as the excellent labels to indicate the content of multiple biomarkers [58,59,60,61]. Our group has come up with a method based on the chip-like GCE composed of three electrodes connected in a parallel for simultaneous detection [43]. Based on a novel redox-active hydrogel containing such metal ions (copper(II) ions, lead(II) ions, and titanium(IV) ions) and sodium alginate (SA) (Figure 2(1)), this analysis of multiple targets (CEA, Neuron specific enolase (NSE), Cytokeratin 19 fragments (CYFRA 21-1)) adopted a series improvement for the GCEs, synchronously obtaining three current peaks located at 0, −0.45 and −0.75 V (vs. Ag/AgCl) (Figure 2(2A)). With this method, they have successfully accomplished the assembly of an immunosensor (Figure 2(2B)) and got a series of proportion relations between tumor markers and responsive current, improving the accurate diagnosis for lung cancer.

Besides, a multiplexed electrochemical immunosensing interface based on metal ions doped chitosan-poly(acrylic acid) nanospheres (CP) was proposed for the simultaneous detection of CEA, CA 19-9, CA 12-5, and CA 24-2 [62]. As depicted in Figure 3, with the crosslinking of glutaraldehyde (GA), four types of metal ions doped CP nanospheres were combined with different Ab_2_ to synthesize the new immuoprobes, and chitosan-Au NPs were used as a common substrate to fix on the Ab_1_. Once sandwich conjugate was formed, four independent potential peaks were produced by copper(II), lead(II), cadmium(II), and zinc(II) ions when the whole system was readout with the Square wave voltammograms (SWV) method, indicating the content of CEA, CA 19-9, CA 12-5, and CA 24-2, respectively. Notably, the preparation of the metal ions doped CP composites was first reported by this work and successfully used as the electrochemical signals. With this design, the proposed electrochemical immunosensor exhibited wide linear ranges towards these four biomarkers: 0.1 to 150 ng mL^−1^ for CEA, 1 to 150 U mL^−1^ for CA 19-9, CA 12-5, and CA 24-2.

In addition, Putnin prepared a sort of dually functional polyethylenimine (PEI)-coated gold nanoparticles as immunoprobes, using the strong electrostatic adsorption to loading of four different types of metal ions, including lead(II) ions, copper(II) ions, cadmium(II) ions, and silver ions [63]. On account of four independent voltammetric signals generated by these four metal ions, which were posited at −0.80, −0.55, −0.20, and +0.05 V, respectively, this multiplexed electrochemical immunosensor can efficaciously detect AFP, CEA, PSA, and interleukin-8 (IL-8).

Metal ions, as the common electrochemical signal substance, have been maturely applied into the labeling of immunosensing interface, especially for the silver ions and copper(II) ions, all can generate sharp and strong electrochemical signal peaks. Besides, metal ions like copper(II) ions also own the excellent catalytic activity to some substrates, for example, hydrogen peroxide.

### 2.3. Metal Nanoparticles

With the advantages of small size, high surface-to-bulk ratio, interesting electric and catalytic properties, metal nanoparticles attract much attention and have been widely used as promising electrochemical signal substances [64,65,66]. Besides, metal nanoparticles as sensing elements can serve as an excellent anchor to fix biomolecules or be directly electrodeposited on immunosensing interfaces to become the electrochemical signal indicator [67,68,69]. Hence, by employing a cascade reaction to form silver nanoparticles, a multiplexed immunosensor array fabricated on a carbon electrode array was prepared with screen-printed technology to simultaneously detect CEA, CA 15-3, and CA 12-5, using the measurement of linear-sweep stripping voltammetric (LSV) [70]. Concretely, pretreated screen-printed carbon electrodes (SPCEs) were covered with the Au NPs/graphene (GR) to serve as the substrate to fix Ab_1_. Once the 3-indoxyl phosphate (3-IP) and silver ions were present in this sensor system, 3-IP was quickly hydrolyzed and produced an indoxyl intermediate which can subsequently reduce silver ions to get a silver deposition. For the synthesis of immunoprobes, many uniform Au NPs were in situ synthesized on the surface of pretreated GR and used to immobilize Ab_2_ and alkaline phosphatase (ALP), finally preparing a catalytic probe. Finally, this electrochemical multiple immunosensor exhibited a limit detection of 0.0015 U mL^−1^ for CA 15-3, 0.00034 U mL^−1^ for CA 12-5, and 0.0012 ng mL^−1^ for CEA.

Besides, Lai and coworkers constructed a multiplexed immunosensing interface based on the SPCEs, using deposited silver nanoparticles as the electrochemical labels to indicate the content of CEA and AFP, respectively (Figure 4) [71]. With the assistance of the chemical coupling of GA, the Ab_1_ of CEA and AFP were fixed on the surface of SPCEs, and sandwich composites were subsequently formed after the process of silver deposition had been accomplished. Thanks to the fact that deposited silver nanoparticles could be directly measured by LSV to generate a sharp and high-signal peak, the proposed multiplexed immunoassay method showed low detection limits down to 3.5 and 3.9 pg mL^−1^, respectively.

For the multiplexed electrochemical immunosensor based on the labeling of metal nanoparticles, the electrochemical signal can be present in a variety of ways, such as nitrification of metal nanoparticles to generate metal ions and the replacement reaction between the inert metal ions and active metal, simultaneously can well serve as the anchors to fix biomolecules.

### 2.4. Prussian Blue

Prussian blue (PB) nanoparticles, an iron-based nanomaterial, also serve as an inherent electrochemical activity [72,73]. Since Neff firstly reported the electrochemical behavior of PB and the successful deposition of a thin layer of PB on a platinum foil in 1978 [74], the wide application of PB has been promoted in various construction of electrochemical immunosensing interface, especially for its development in multiplexed electrochemical immunoassay [75,76]. Moreover, for the modification ways of PB on an electrode surface, there are two main sorts of common methods, including post-modification [77,78] and in-situ preparation [79,80], and both of them can generate the legible signal peaks. For example, Lai and coworkers reported on an ultrasensitive multiplexed electrochemical immunosensing interface to simultaneously detect CEA and AFP [81]. By developing a dual signal amplification tracer based on glucose oxidase (GOD)-functionalized carbon nanotubes, the PB-mediated electron transfer process can be effectively accelerated to improve the detection performance of the whole immunoassay array (Figure 5(2)). To prepare this signal substrate, they synthesized uniform and nanosized PB nanoparticles in a PDDA and chitosan (1% HAc solution) and directly dropped them on the surface of SPCEs (Figure 5(1)). With a DPV measurement, PB in this signal substrate was well readout and generated an excellent signal peak at about +0.05V. Interestingly, this immunoassay array can complete the analysis of multiple biomarkers by selecting only one electrochemical signal substance as a label, avoiding the fussy labeling of different electrochemical signal substances.

The PB-constructed signal substrate can also be used in preparing a label-free microfluidic paper-based electrochemical aptasensor. Wang et al. successfully fabricated a paper device with microfluidic channels while a three-electrode system was made by the screen-printing method [82]. As illustrated in Figure 6, two sorts of working electrodes were modified by different substrate materials (amino-functional graphene-thionine and PB-poly(3,4-ethylenedioxythiophene) (PEDOT)), detecting CEA and NSE in a clinical sample, respectively. Just for the addition of PEDOT, it not only facilitated electron transfer capability during the readout of electrochemical signal but also increased the amount of immobilized aptamers, thus increasing the sensitivity of the electrochemical immunosensing interface for detecting NSE.

Prussian blue, whether it is post-synthesis or in-situ preparation, can all generate an electrochemical signal. The rise of its derivatives has promoted the sensitive detection of multiple tumor markers based on the multiplexed immunosensing interface.

### 2.5. Other Substance for Electrochemical Signal Generation

As a substitute for biozyme, some artificial nanozymes have been widely applied to the designing of multiplexed electrochemical immunosensing interfaces due to their considerable environmental tolerance, easy storage, and ideal catalytic activity [83,84,85]. Except for the aforementioned ways in which electrochemical signals are generated by the redox of electroactive species, nanozymes with peroxidase-like activity can also catalyze the H_2_O_2_ to generate current variation with an i-t measurement [86,87]. Employing the platinum nanoparticle-functionalized mesoporous silica (Pt@SBA-15) to form an immunosensor array has been successfully used for the simultaneous determination of different analytes. Ma and coworkers used Pt@SBA-15 as a label to improve the limited stability of the process in which biozyme is attached to the Ab_2_ with a time-consuming, costly labeling process that often leads to the denaturization of the biomolecules [88]. Benefiting from this label, this multiplexed electrochemical immunosensing interface showed low detection limits for diethylstilbestrol (0.28 pg mL^−1^) and estradiol (1.2 pg mL^−1^).

Besides the aforementioned Pt nanozymes with peroxidase-like activity, Sun and coworkers proposed a multiplexed enzyme-free electrochemical immunosensor based on the ZnO nanorods modified reduced graphene oxide-paper electrode, using the catalytic amplification strategy induced by BSA-stabilized silver nanoparticles (Figure 7) [89]. Because of the existence of silver nanoparticles, H_2_O_2_ can be well reduced to induce a current response. With this strategy, the proposed multiplexed immunosensor exhibited a wide linear range of 0.002–120 mIU mL^−1^ for human chorionic gonadotropin (HCG), 0.001–110 ng mL^−1^ for PSA, and 0.001–100 ng mL^−1^ for CEA.

In consideration of some nanozymes prepared with noble metals like Pt and Pd; it is necessary for future studies to develop artificial nanozymes based on the advantages of low-cost, high catalysis activity. Catalytic-amplification system is an excellent mediator to sensitively detect multiple tumor markers and deserves to be promoted in more analysis fields, like inorganic molecules detection.

## 3. Conclusions and Future Prospects

Overall, we have summarized recent advances in the labeling of electrochemical signal molecules for the analysis of multiple tumor markers. We fully introduced the characterizations of every signal material, including organic molecules, metal ions, metal nanoparticles, Prussian blue, and other substances for signal generation, and discussed their concrete application (such as the participation of immunoprobe synthesis, substrate construction) in designing electrochemical immunosensing interface (Table 1). However, for the labeling of electrochemical signal substances for multiplexed immunosensors, there are some challenges that still need to be solved. (1) Existing fixation methods of signal substance always have the deficiencies, such as the unstable physisorption to cause leakage, fussy covalent bonding to lead time-consuming. Hence, developing good stability and a time-saving labeling method is necessary for improving the readout of the electrochemical signal method. (2) Considering the pollution problem of organic molecules or heavy metal ions (such as methylene blue, thionine, and lead ions, etc.), it is recommended to design a signal attenuation interface (it can effectively degrade the harmful signal substance to reduce electrochemical signal) or reproducible interface (it can reuse these signal labels).

## Figures and Tables

**Figure 1 molecules-27-00267-f001:**
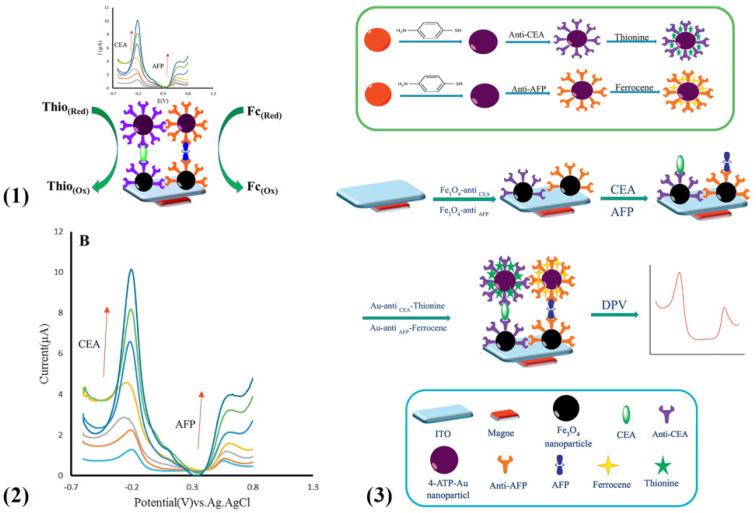
(**1**) Sandwich-type electrochemical immunosensor for the simultaneous detection of CEA and AFP. (**2**) DPV responses of the immunosensor incubated with a series of different concentrations of CEA and AFP. (**3**) Assembly schematic of electrochemical immunosensor for the detection of CEA and AFP (reprinted with permission from Ref. [54], Copyright 2018, Elsevier).

**Figure 2 molecules-27-00267-f002:**
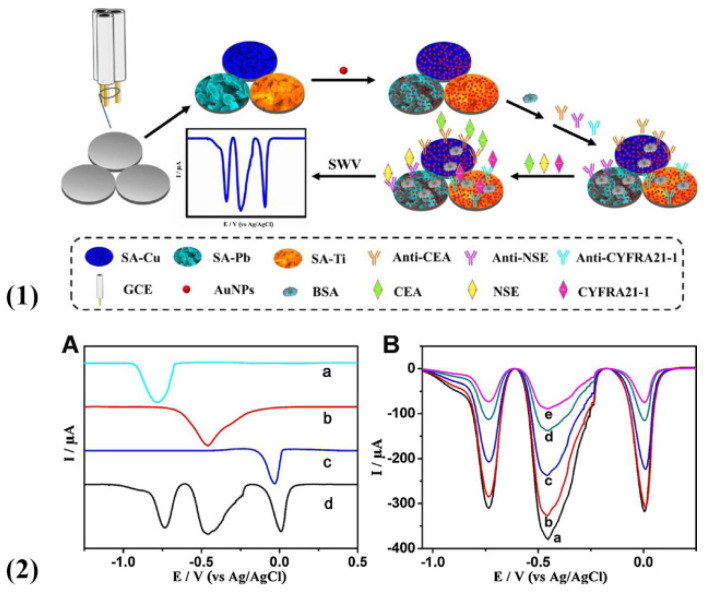
(**1**) Schematic for the fabrication process of the immunosensing interface. (**2**) Typical SWV signals of SA−Cu (a), SA−Pb (b), SA−Ti (c), and a mixture of these three hydrogels (d) (**A**). SWV characterizations of the modified procedure of electrodes (**B**) (reprinted with permission from Ref. [43], Copyright 2018, Elsevier).

**Figure 3 molecules-27-00267-f003:**
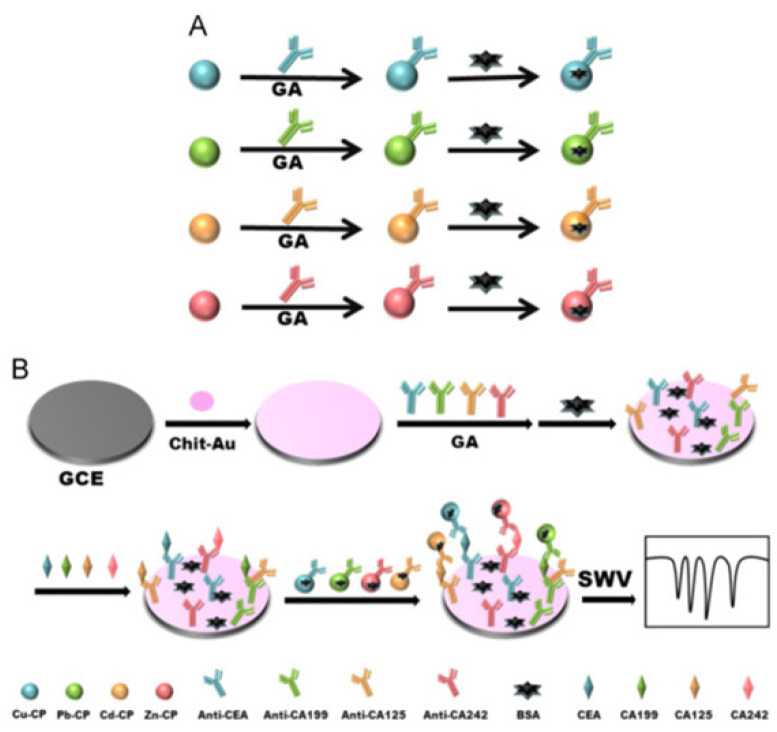
Preparation of various immunoprobes based on metal ions doped chitosan-poly(acrylic acid) nanospheres (**A**). Schematic representation for the electrochemical multiplexed immunosensor of CEA, CA 19-9, CA 12-5, and CA 24-2 (**B**) (reprinted with permission from Ref. [62], Copyright 2016, Elsevier).

**Figure 4 molecules-27-00267-f004:**
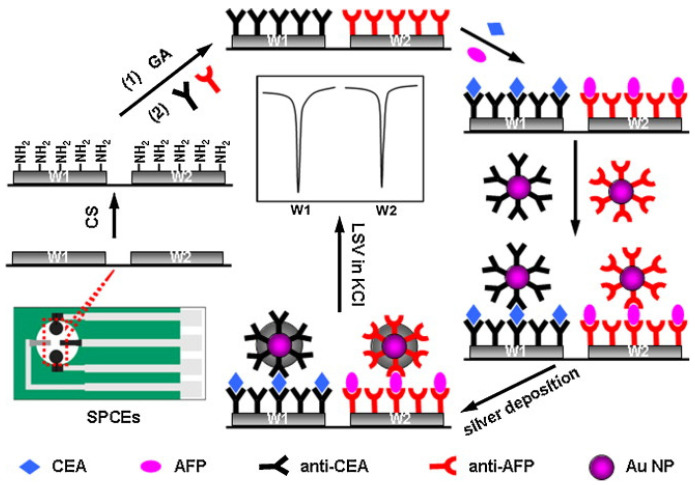
Schematic of the electrochemical immunosensor array and detection strategy by linear-sweep stripping voltammetric analysis of Ag NPs catalytically deposited on the immunosensor surface by gold nanolabels (reprinted with permission from Ref. [71], Copyright 2012, Elsevier).

**Figure 5 molecules-27-00267-f005:**
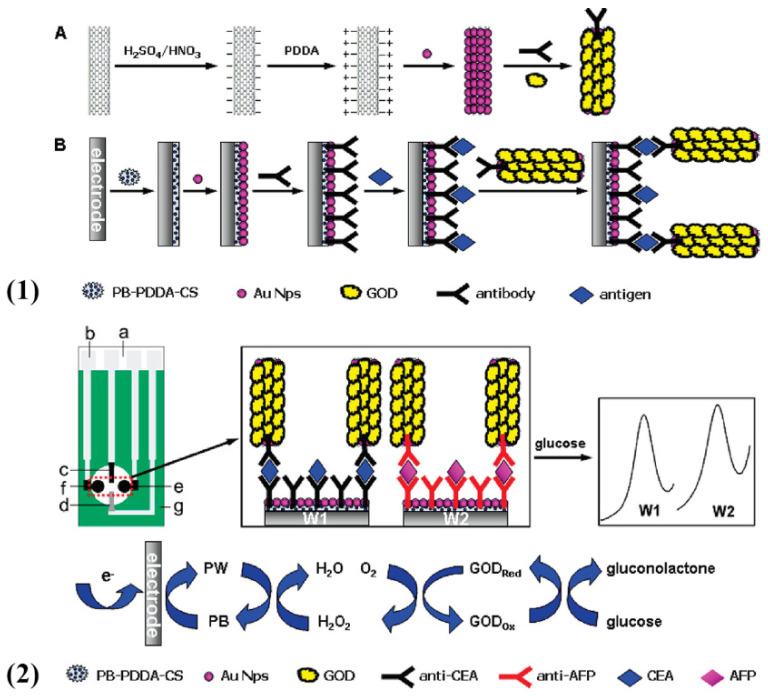
(**1**) Schematic representation of the preparation of immunosensors (A) and sandwich-type electrochemical immunoassay (B). (**2**) Schematic representation of a multiplexed electrochemical immunoassay with an immunosensor array and electrochemical response mechanism (reprinted with permission from Ref. [81], Copyright 2009, American Chemical Society).

**Figure 6 molecules-27-00267-f006:**
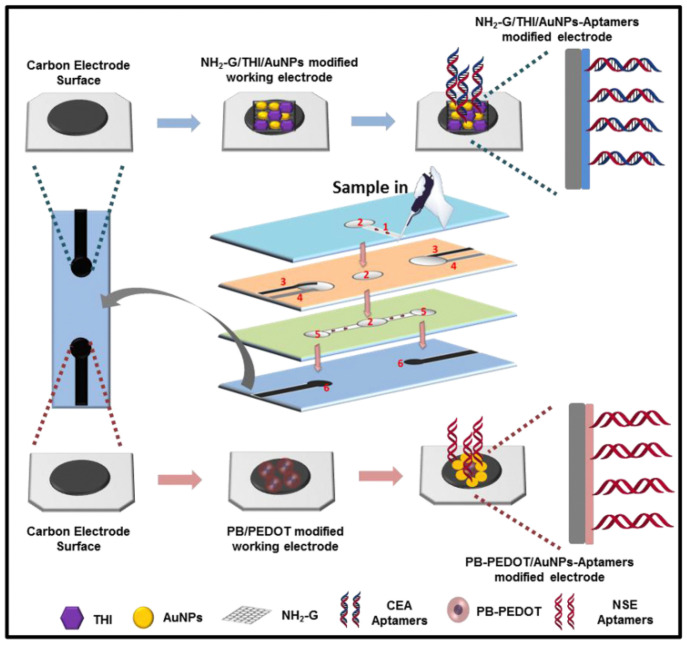
Fabrication and modification process of the multi-parameter electrochemical paper-based aptasensor (reprinted with permission from Ref. [82], Copyright 2019, Elsevier).

**Figure 7 molecules-27-00267-f007:**
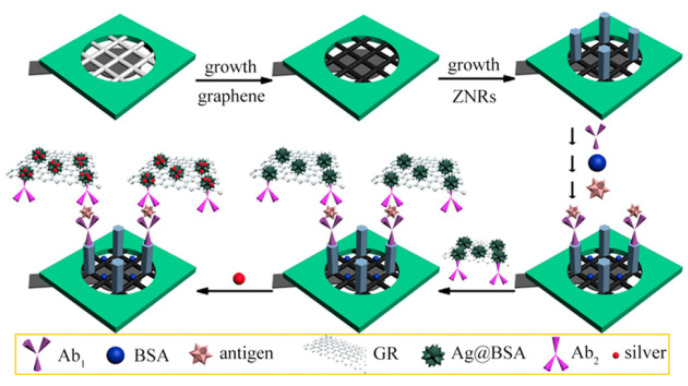
The fabrication process of the multiplexed enzyme-free electrochemical immunosensor based on the BSA-stabilized silver nanoparticles (reprinted with permission from Ref. [89], Copyright 2015, Elsevier).

**Table 1 molecules-27-00267-t001:** Summary of the advantages and limitations for various electrochemical signal substances and some reported works.

Species	Advantages	Limitations	Electrochemical Signal Substance	Detection Object	Ref.
Organic molecules	Easy accessibility, Simple loading, Excellent redox spikes	Unstable physisorption to cause leakage, Fussy covalent bonding to lead time-consuming, Environmental contamination	Methylene blue, Thionine	CEA, AFP	[53]
			Thionine, Ferrocene	CEA, AFP	[54]
			Anthraquinone 2-carboxylic acid, Thionine, Tris(2,2(-bipyridine-4,4(-dicarboxylic acid) cobalt(III), Ferrocene	CEA, CA 19-9, CA 12-5, CA 24-2	[55]
Metal ions	Easy accessibility, Sharp electrochemical signal peaks	Consideration of heavy metal ions contamination	Copper(II) ions, Lead(II) ions, Titanium(IV) ions	CEA, NSE, CYFRA 21-1	[43]
			Copper(II) ions, Lead(II) ions, Cadmium(II) ions, zinc(II) ions	CEA, CA 19-9, CA 12-5, CA24-2	[62]
			Copper(II) ions, Lead(II) ions, Cadmium(II) ions, Silver ions	CEA, AFP, PSA, IL-8	[63]
Metal nanoparticles	Small size, High surface-to-bulk ratio, High catalytic activity	Easily oxidation for a long period of storage	Silver nanoparticles	CEA, CA 12-5, CA 15-3	[70]
			Silver nanoparticles	CEA, AFP	[71]
Prussian blue	High catalytic activity, Excellent electrochemical signal peaks	Requirement of pre-preparation	PB-PDDA-CS	CEA, AFP	[81]
			PB-PEDOT	CEA, NSE	[82]
Other substances	Excellent catalytic system Needlessness of signal molecules labeling	Common usage of noble metal-based nanozyme	Pt NPs-functionalized mesoporous silica	Diethylstilbestrol Estradiol	[88]
			BSA-stabilized silver nanoparticles	CEA, PSA, HCG	[89]

## Data Availability

No new data were created or analyzed in this study. Data sharing is not applicable to this article.

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
