# Peer review of "Electrochemical Signal Substance for Multiplexed Immunosensing Interface Construction: A Mini Review"

_molecules, 2022, doi:10.3390/molecules27010267_

Round 1

Reviewer 1 Report

Though authors have compiled reports on Organic molecules, Metal ions, Metal Nanoparticles and Prussian blue NPs as electrochemical signal substances for multiplexed immunosensing there are other hybrid nanosystem recently demonstrated in electrochemical immunosensing of clinical important biomarkers. For example if authors could provide a table of metal-carbon, metalloid-polymer composite and electroadsorbed redox-active system for immunosensing under section 2.5 it would benefit more readers for a better reference. Moreover, electrodeposition based method merits should be mentioned in appropriate places by which issues related to physisorption on sensor substrate could be addressed.

Author Response

Response to Reviewer #1

Q1: Though authors have compiled reports on Organic molecules, Metal ions, Metal Nanoparticles and Prussian blue NPs as electrochemical signal substances for multiplexed immunosensing there are other hybrid nanosystem recently demonstrated in electrochemical immunosensing of clinical important biomarkers. For example if authors could provide a table of metal-carbon, metalloid-polymer composite and electroadsorbed redox-active system for immunosensing under section 2.5 it would benefit more readers for a better reference. Moreover, electrodeposition based method merits should be mentioned in appropriate places by which issues related to physisorption on sensor substrate could be addressed.

Response: Thank you very much for your suggestion. The works reported on the electrochemical signal substance like metal-carbon, metalloid-polymer composites and electroadsorbed redox-active system are mostly used for the single tumor marker analysis instead of the multiplexed analysis system. The novel electrochemical signal generation system you provided can be a future direction for the excellent construction of multiplexed immunosensing interface. Alternatively, we summarized a table of various electrochemical signal substance used in designing multiplexed electrochemical immunosensing interface on page 8 as follows.

      Besides, electrodeposition based method merits was mentioned in the section 2.3 on the page 5 as follows.

      “…Besides, metal nanoparticles as sensing elements can serve as an excellent anchor to fix biomolecules or be directly electrodeposited on immunosensing interface to become the electrochemical signal indicator[67, 68, 69]…”

Table 1. Summary of the advantages, and limitations for various electrochemical signal substance, and some reported works.

Species

Advantages

Limitations

Electrochemical signal substance

Detection object

Ref.

Organic molecules

Easy accessibility,

Simple loading, Excellent redox spikes

Unstable physisorption to cause leakage,

Fussy covalent bonding to lead time-consuming,

Environmental contamination

Methylene blue,

Thionine

CEA, AFP

[53]

Thionine,

Ferrocene

CEA, AFP

[54]

Anthraquinone 2-carboxylic acid,

Thionine,

Tris(2,2(-bipyridine-4,4(-dicarboxylic acid) cobalt(III),

Ferrocene

CEA,

CA 19-9,

CA 12-5,

CA 24-2

[55]

Metal ions

Easy accessibility,

Sharp electrochemical signal peaks

Consideration of heavy metal ions contamination

Copper(II) ions,

Lead(II) ions,

Titanium(IV) ions

CEA, NSE,

CYFRA 21-1

[43]

Copper(II) ions,

Lead(II) ions,

Cadmium(II) ions,

zinc(II) ions

CEA,

CA 19-9,

CA 12-5,

CA24-2

[62]

Copper(II) ions,

Lead(II) ions,

Cadmium(II) ions, Silver ions

CEA, AFP,

PSA, IL-8

[63]

Metal nanoparticles

Small size,

High surface-to-bulk ratio,

High catalytic activity

Easily oxidation for a long period storage

Silver nanoparticles

CEA,

CA 12-5,

CA 15-3

[70]

Silver nanoparticles

CEA, AFP

[71]

Prussian blue

High catalytic activity,

Excellent electrochemical signal peaks

Requirement of pre-preparation

PB-PDDA-CS

CEA, AFP

[81]

PB-PEDOT

CEA, NSE

[82]

Other substances

Excellent catalytic system

Needlessness of signal molecules labeling

Common usage of noble metal based nanozyme

Pt NPs-functionalized mesoporous silica

Diethylstilbestrol

Estradiol

[88]

BSA-stabilized silver nanoparticles

CEA, PSA, HCG

[89]

Reviewer 2 Report

The submitted Manuscript entitled “Electrochemical signal substance for multiplexed immunosensing interface construction: A mini review” is well written but need some improvement before accept in this esteemed Journal. As per our opinion this review may be accepted after minor revision. Comments for manuscript are as follows:

  1. A table should be incorporated, in which summarize the steps taken in the field of multiplexed immunosensing interface construction by different signal substrate such as organic molecules, metal ions, metal nanoparticles, Prussian blue and other substances for electrochemical signal generation along with its advantages and limitations.
  2. In last of each section i.e. section 2.1 to 2.5, it should be summarized and discuss which molecules/material/labeling is excellent.

Author Response

Response to Reviewer #2

Q1: A table should be incorporated, in which summarize the steps taken in the field of multiplexed immunosensing interface construction by different signal substrate such as organic molecules, metal ions, metal nanoparticles, Prussian blue and other substances for electrochemical signal generation along with its advantages and limitations.

Response: Thank you very much for your suggestion. A table of various electrochemical signal substance used in designing multiplexed electrochemical immunosensing interface has been provided on page 8 as follows.

Species

Advantages

Limitations

Electrochemical signal substance

Detection object

Ref.

Organic molecules

Easy accessibility,

Simple loading, Excellent redox spikes

Unstable physisorption to cause leakage,

Fussy covalent bonding to lead time-consuming,

Environmental contamination

Methylene blue,

Thionine

CEA, AFP

[53]

Thionine,

Ferrocene

CEA, AFP

[54]

Anthraquinone 2-carboxylic acid,

Thionine,

Tris(2,2(-bipyridine-4,4(-dicarboxylic acid) cobalt(III),

Ferrocene

CEA,

CA 19-9,

CA 12-5,

CA 24-2

[55]

Metal ions

Easy accessibility,

Sharp electrochemical signal peaks

Consideration of heavy metal ions contamination

Copper(II) ions,

Lead(II) ions,

Titanium(IV) ions

CEA, NSE,

CYFRA 21-1

[43]

Copper(II) ions,

Lead(II) ions,

Cadmium(II) ions,

zinc(II) ions

CEA,

CA 19-9,

CA 12-5,

CA24-2

[62]

Copper(II) ions,

Lead(II) ions,

Cadmium(II) ions, Silver ions

CEA, AFP,

PSA, IL-8

[63]

Metal nanoparticles

Small size,

High surface-to-bulk ratio,

High catalytic activity

Easily oxidation for a long period storage

Silver nanoparticles

CEA,

CA 12-5,

CA 15-3

[70]

Silver nanoparticles

CEA, AFP

[71]

Prussian blue

High catalytic activity,

Excellent electrochemical signal peaks

Requirement of pre-preparation

PB-PDDA-CS

CEA, AFP

[81]

PB-PEDOT

CEA, NSE

[82]

Other substances

Excellent catalytic system

Needlessness of signal molecules labeling

Common usage of noble metal based nanozyme

Pt NPs-functionalized mesoporous silica

Diethylstilbestrol

Estradiol

[88]

BSA-stabilized silver nanoparticles

CEA, PSA, HCG

[89]

Q2: In last of each section i.e. section 2.1 to 2.5, it should be summarized and discuss which molecules/material/labeling is excellent.

Response: Thanks for your careful review. The excellent molecules/material/labeling have been summarized and discussed in last of each section in the revised manuscript.

     2.1 Organic molecules (on page 3)

“…Based on the labeling of organic molecules, the multiplexed immunosensor all exhibited considerable electrochemical signal, which can be independently present at corresponding position. Especially for the organic dyestuff, they can generate high current value even if it was the trace…”

     2.2 Metal ions (in lines 39-42 on page 5)

“…Metal ions, as the common electrochemical signal substance, have been maturely applied into the labeling of immunosensing interface, especially for the silver ions and copper(II) ions, all can generate the sharp and strong electrochemical signal peaks. Besides, metal ions like copper(II) ions also own the excellent catalytic activity to some substrate, for example, hydrogen peroxide…”

     2.3 Metal nanoparticles (in lines 27-30 on pages 6)

“…For the multiplexed electrochemical immunosensor based on the labeling of metal nanoparticles, the electrochemical signal can be present in a variety of ways, such as, nitrification of metal nanoparticles to generate metal ions, and the replacement reaction between the inert metal ions and active metal, simultaneously can well serve as the anchors to fix biomolecules…”

     2.4 Prussian blue (on pages 7)

“…Prussian blue, whether it is post-synthesis or in-situ preparation, they all can generate electrochemical signal. The rise of its derivatives has promote the sensitive detection of multiple tumor markers based on the multiplexed immunosensing interface.”

     2.5 Other substance for electrochemical signal generation (on pages 9)

     “…Consideration of some nanozymes is prepared by noble metal like Pt, Pd, it is necessary for the future study to develop the artificial nanozymes based on the advantages of low-cost, high catalysis activity. Catalytic-amplification system is an excellent mediator to sensitively detect multiple tumor markers, and deserved to be promoted in more analysis field, like inorganic molecules detection…”

Reviewer 3 Report

General comment:

The authors make a systematic contribution to the research literature related to electrochemical signal substance for multiplexed immunosensing interface construction. Further, they tried to provide useful informative data on extend and severity of electrochemical signal substance. This manuscript is based on a comprehensive data set and the authors have thoroughly designed this review. This manuscript could be an important reference for future studies. However, a minor revision is still needed to improve the quality of this paper. Please revise the manuscript to address the expressed concerns.

Abstract:

- The abstract needs improvement.  An abstract should start with the motivation (i.e., hypotheses or questions), summarize principal facts and conclusions of your paper, and always end with the significance of the findings so that the reader knows how your work fits in the broader scope of the literature and can identify your contribution. The first sentence can be said about any study related to health literacy is not sufficient scientific justification for a study.  Also, the last sentence is insufficient because it could be said about every study ever performed.  What is needed is a statement as to what new science you are adding.

Introduction:

In this manuscript, although, the introduction is relevant and theory based, but need improvement:

Regarding your review of the literature what is already known on the topic, and the gaps in the literature that this research aimed to fill? I recommend used the following literatures to update the introduction: Dual-signaling electrochemical ratiometric strategy for simultaneous quantification of anticancer drugs/ Highly sensitive electrochemical immunosensor for the simultaneous detection of multiple tumor markers for signal amplification

  • The aim of the study was stated at the end of the introduction. The aim throughout should be to excite and interest, not bore, the reader, and answer the question: “Why was this work embarked upon?”

Discussion:

- Discussion is general information with repetition of results, without saying much that is new science, or contributes to the broader understanding of electrochemical signal substance.  It needs a significant refocus to show what the new scientific points add to science. 

- Further the authors could not make an effective comparison with other research literature in study.

- What is the limitation for this research?

- What is the practice implications and strength of your finding?

Conclusion:

- Finally, the conclusions should be one paragraph, not a repetition of what I have already read, and explain the significance of your findings, which later becomes the last sentence of the abstract.

Author Response

Response to Reviewer #3

Q1: Abstract: The abstract needs improvement. An abstract should start with the motivation (i.e., hypotheses or questions), summarize principal facts and conclusions of your paper, and always end with the significance of the findings so that the reader knows how your work fits in the broader scope of the literature and can identify your contribution. The first sentence can be said about any study related to health literacy is not sufficient scientific justification for a study. Also, the last sentence is insufficient because it could be said about every study ever performed. What is needed is a statement as to what new science you are adding.

Response: Thanks for your careful review. According to your suggestion, the abstract has been improved on page 1 as follows.

     Appropriate labeling method of signal substance is necessary for the construction of multiplexed electrochemical immunosensing interface to enhance the specificity for the diagnosis of cancer. So far, various electrochemical substance including organic molecules, metal ions, metal nanoparticles, Prussian blue, and other methods for electrochemical signal generation have been successfully applied in multiplexed biosensor designing. However, few works have been reported on the summary of electrochemical signal substance applied in constructing multiplexed immunosensing interface. Herein, according to the classification of labeled electrochemical signal substance, this review has summarized the recent state-of-art development for the designing of electrochemical immunosensing interface for simultaneous detection of multiple tumor markers. After that, conclusion and prospects for future applicaiton of electrochemical signal substance in multiplexed immunosensors are also discussed. The current review can provide the comprehensive summary of signal substance selection for workers researched in electrochemical sensor, and further make contributions for the designing of multiplexed electrochemical immunosensing interface with well signal.

Q2: Introduction: In this manuscript, although, the introduction is relevant and theory based, but need improvement:

Regarding your review of the literature what is already known on the topic, and the gaps in the literature that this research aimed to fill? I recommend used the following literatures to update the introduction: Dual-signaling electrochemical ratiometric strategy for simultaneous quantification of anticancer drugs/Highly sensitive electrochemical immunosensor for the simultaneous detection of multiple tumor markers for signal amplification.

The aim of the study was stated at the end of the introduction. The aim throughout should be to excite and interest, not bore, the reader, and answer the question: “Why was this work embarked upon?”

Response: Thank you very much for your suggestion. This work is aimed to summarize the reported signal substance used in multiplexed electrochemical immunosensor, and help the researchers designing excellent immunosensing interface with well signal in the future. This purpose has been stated at the end of the introduction on page 2 in the revised manuscript. Besides, according to your suggestion, we have also updated the introduction at the perspective of sensitivity amplification for the multiplexed electrochemical immunosensor on page 1 in the revised manuscript.

    “…Sensitivity, as the crucial parameters for evaluation of electrochemical immunosnesor, is always defined as the signal variation caused by the incubation of antigen per unit concentration [32, 33, 34, 35]. High sensitivity is beneficial for accurate detection of multiple tumor markers [36, 37, 38]. Labeling of electrochemical signal substance, a necessary step in the construction of electrochemical immunosening interface [39, 40], is closely associated with the final readout of electrochemical signal to influence the sensitivity of immunosensor [41, 42].…”

    “…This timely review is aimed to provide a table of reported signal substance used in multiplexed electrochemical immunosensor, and help researchers designing excellent immunosensing interface with well signal in the future.”

Q3: Discussion: Discussion is general information with repetition of results, without saying much that is new science, or contributes to the broader understanding of electrochemical signal substance. It needs a significant refocus to show what the new scientific points add to science.

Further the authors could not make an effective comparison with other research literature in study.

What is the limitation for this research?

What is the practice implications and strength of your finding?

Response: Thanks for your careful review. This new research focused on providing references for labeling and selection of signal substances for the electrochemical analysis of multiple tumor markers. We have fully summarized the reported works based on various electrochemical signal. It will obviously promote the designing of multiplexed electrochemical immunosensing interface. In future works, we plan to improve the significance of this research by summarizing more multiplexed analyte detection like inorganic molecules (Uric acid, dopamine) and nucleic acid.

Q4: Conclusion: Finally, the conclusions should be one paragraph, not a repetition of what I have already read, and explain the significance of your findings, which later becomes the last sentence of the abstract.

Response: Thanks for your careful review. The conclusion has been rewritten in one paragraph on page 9 in the revised manuscript, concluding the application of various electrochemical signal in multiplexed immunosensors, and elaborating the subsistent problems about the labeling process of the signal substamce.

    “Overall, we have summarized recent advance in the labeling of electrochemical signal molecules for the analysis of multiple tumor marker. We fully introduced the characterizations of every signal materials including organic molecules, metal ions, metal nanoparticles, Prussian blue and other substance for signal generation, and discuss their concrete application (such as the participation of immunoprobe synthesis, substrate construction) in designing electrochemical immunosensing interface (Table 1). However, for the labeling of electrochemical signal substance for multiplexed immunosensor, there are some challenges that still need to be solved. (1) Existing fixation methods of signal substance always have the deficiencies, such as the unstable physisorption to cause leakage, fussy covalent bonding to lead time-consuming. Hence, developing good stability and time-saving labeling method is necessary for improving the readout of electrochemical signal method. (2) Considering the pollution problem of organic molecules or heavy metal ions, (such as methylene blue, thionine and lead ions, etc) it is recommended to design signal attenuation interface (it can effectively degrade the harmful signal substance to reduce electrochemical signal) or reproducible interface (it can reuse these signal labels).”
